# Stem Cell-Induced Inflammation in Cholesteatoma Is Inhibited by the TLR4 Antagonist LPS-RS

**DOI:** 10.3390/cells9010199

**Published:** 2020-01-14

**Authors:** Matthias Schürmann, Johannes F. W. Greiner, Verena Volland-Thurn, Felix Oppel, Christian Kaltschmidt, Holger Sudhoff, Barbara Kaltschmidt

**Affiliations:** 1Department of Otolaryngology, Head and Neck Surgery, Klinikum Bielefeld, 33604 Bielefeld, Germany; MATTHIAS.SCHUERMANN@klinikumbielefeld.de (M.S.); Verena-Volland-Thurn@gmx.de (V.V.-T.); Felix.Oppel@klinikumbielefeld.de (F.O.); holger.sudhoff@rub.de (H.S.); 2Department of Cell Biology, University of Bielefeld, 33619 Bielefeld, Germany; johannes.greiner@uni-bielefeld.de (J.F.W.G.); c.kaltschmidt@uni-bielefeld.de (C.K.); 3AG Molecular Neurobiology, University of Bielefeld, 33619 Bielefeld, Germany

**Keywords:** cholesteatoma, stem cells, inflammation, TRL4, NF-κB, LPS-RS, IL-6

## Abstract

Cholesteatoma is a severe non-cancerous lesion of the middle ear characterized by massive inflammation, tissue destruction, and an abnormal growth of keratinized squamous epithelium. We recently demonstrated the presence of pathogenic stem cells within cholesteatoma tissue, unfortunately their potential roles in regulating disease-specific chronic inflammation remain poorly understood. In the presented study, we utilized our established human in vitro cholesteatoma stem cell model for treatments with lipopolysaccharides (LPS), tumor necrosis factor α (TNFα), and the TLR4-antagonist LPS from *R. sphaeroides* (LPS-RS) followed by qPCR, western blot, and immunocytochemistry. Middle ear cholesteatoma stem cells (ME-CSCs) showed a significantly increased expression of TLR4 accompanied by a significantly enhanced LPS-dependent pro-inflammatory gene expression pattern of TNFα, IL-1α, IL-1ß, IL-6, and IL-8 compared to non-pathogenic control cells. LPS-dependent pro-inflammatory gene expression in ME-CSCs was driven by an enhanced activity of NF-κB p65 leading to a TNFα-mediated feed-forward-loop of pro-inflammatory NF-κB target gene expression. Functional inactivation of TLR4 via the TLR4-antagonist LPS-RS blocked chronic inflammation in ME-CSCs, resulting in a nearly complete loss of IL-1ß, IL-6, and TNFα expression. In summary, we determined that ME-CSCs mediate the inflammatory environment of cholesteatoma via TLR4-mediated NF-κB-signaling, suggesting a distinct role of ME-CSCs as drivers of cholesteatoma progression and TLR4 on ME-CSCs as a therapeutic target.

## 1. Introduction

Cholesteatoma is a potentially life-threatening inflammatory lesion of abnormally growing keratinizing squamous epithelium in the middle ear (Figure 1A) resulting in clinical symptoms like hearing loss, ear discharge, and ear pain [1]. The non-cancerous but intensely proliferating squamous epithelium can also locally invade and destroy nearby structures like the temporal bone and the auditory ossicles [2]. Severe complications of cholesteatoma may further include sigmoid sinus thrombosis, epidural abscess, encephalitis, or meningitis. Approximately 9.2 new cholesteatoma cases in 100,000 people are reported per year in northern Europe [1]. Medical management of cholesteatoma is challenging (reviewed in [3]) and most often limited to surgical removal of cholesteatoma. Cholesteatoma can be classified into acquired and congenital cholesteatoma [4], with congenital cholesteatoma representing 2–4% of all cases and occurring only in children between the ages of 4–6 years [5]. In contrast, acquired cholesteatoma affects both children and adults and is closely associated with long lasting inflammation and infection of the middle ear, e.g., chronic otitis media [2,4]. Even though there are different existing theories regarding the formation of acquired cholesteatoma (reviewed in [3,6]), massive inflammation is one of its major characteristics and is also discussed as a pre-requisite for cholesteatoma formation and recurrence [7]. Hence the presence of inflammatory mediators in cholesteatoma tissue was studied and exhibits a higher presence of the cytokines TNFα, IL-6, IL-8, IL-1α, and IL-1ß in comparison to non-inflamed external auditory canal skin tissue [8,9]. Bone resorption and destruction was also shown to be positively correlated to inflammation, the presence of bacterial lipopolysaccharides (LPS), and expression of inflammatory mediators like TNFα, IL-1α, and IL-6 in cholesteatoma [10,11,12]. In addition, an enhanced expression of the Toll-like receptor 4 (TLR4) was observed in cholesteatoma tissue compared to external auditory canal skin tissue [13]. Interestingly, in a mouse model, TLR4 was demonstrated to promote not only local inflammation but also bone destruction [14]. Notably, the activation of the transcription factor NF-κB was shown to be elevated in cholesteatoma tissue [15,16]. NF-κB is not only recognized to mediate cell survival, but is also known as a key regulator in the pathogenesis of inflammatory diseases [17]. For example, the hypertrophy of nasal mucosa was linked to enhanced activity of NF-κB [18] in chronic rhinosinusitis exhibiting overexpression of TLR4 [19]. However, the cellular mediators potentially linking NF-κB-signaling and the pro-inflammatory environment of cholesteatoma still remain unknown.

In the present study, we aimed to determine these cellular mediators of the highly pro-inflammatory environment in cholesteatoma. We recently demonstrated the presence of stem cells contributing to cholesteatoma pathogenesis within cholesteatoma tissue [20]. Middle ear cholesteatoma derived stem cells (ME-CSCs) and stem cells isolated from auditory canal skin (ACSCs), which were utilized as control cells, were successfully isolated and cultivated in vitro, where they showed the ability for spheres formation, self-renewal, and multipotent differentiation. Exposure of factors mimicking the microenvironment of cholesteatoma further resulted in differentiation of ME-CSCs into keratinocyte-like cells, suggesting a potential novel disease mechanism [20]. Extending these promising findings, our present observations demonstrate a significantly increased expression of TRL4 accompanied by a significantly enhanced LPS-dependent pro-inflammatory gene expression pattern in ME-CSCs compared to ACSCs. ME-CSCs treated with LPS further showed an elevated activity of NF-κB p65 in comparison to ACSCs, promoting a TNFα-mediated feed-forward-loop of chronic NF-κB target gene expression. Interestingly, the TNFα response to this feed-forward-loop is specifically more sensitive in ME-CSCs. As a clinical perspective, we functionally blocked the TLR4 using the TLR4-antagonist LPS from *R. sphaeroides* (LPS-RS), resulting in a nearly complete loss of IL-1α, IL-1ß, IL-6, and TNFα expression in in ME-CSCs. In summary, we determined that ME-CSCs regulate the inflammatory environment of cholesteatoma via TLR4-mediated NF-κB-signaling, suggesting a distinct role of ME-CSCs as drivers of cholesteatoma progression in an inflammation-dependent manner.

## 2. Materials and Methods

### 2.1. Ethics Statement and Human Samples

Acquired cholesteatomas and external auditory canal skin specimens were obtained from patients undergoing middle ear surgery at Klinikum Bielefeld Mitte (Bielefeld, Germany). Fully informed written consent was obtained prior to surgery and all clinical investigations were ethically approved (Reg. no. 2235) and conducted according to the principles of the Declaration of Helsinki (1964) and local guidelines (Bezirksregierung Detmold/Münster). The removed tissue samples were used for isolation of stem cells and paraffin sectioning.

### 2.2. Isolation and Culture of Cholesteatoma and Auditory Canal Skin Stem Cells

Middle ear cholesteatoma stem cells (ME-CSCs) and auditory canal skin stem cells (ACSCs) were isolated and cultivated with addition of 10% human blood plasma or as free-floating spheres according to our previously described protocol [20]. Briefly, tissue samples were digested with Collagenase I (0.375 U/mL in PBS, SERVA Electrophoresis GmbH, Heidelberg, Germany) for at least 1 h and mechanically disintegrated followed by stem cell isolation at 37 °C and 5% CO2 in standard medium comprising DMEM/F-12 (Sigma-Aldrich, Merck KGaA, Darmstadt, Germany), L-Glutamin (200 mM, Sigma Aldrich), penicillin (10 U/mL, Sigma Aldrich), streptomycin (10 U/mL, Sigma Aldrich), amphotericin B (25 µg/mL, Sigma Aldrich), EGF (20 ng/mL, Peprotech, Hamburg, Germany), bFGF (40 ng/mL, Peprotech), and B27 supplement (Gibco, Thermo Fisher Scientific, Waltham, MA, United States). This medium was either supplemented with 10% human blood plasma for efficient expansion according to [21] or heparin (2 μg/mL, Sigma-Aldrich), to allow sphere formation.

### 2.3. Haematoxylin and Eosin Staining of Cryostat Sections

Frozen 10 μm thick paraffin sections of cholesteatoma tissue and external auditory canal skin were subjected to H&E staining followed by microscopically examination.

### 2.4. Treatment of ME-CSCs and ACSCs with LPS, Heat-Killed Bacteria, TNFα, or LPS-RS

ME-CSCs and ACSCs were seeded in 6-well plates (CytoOne^®^ Multiple Well Plates, STARLAB GmbH, Hamburg, Germany, 5.3 x 10^3^ cells/cm^2^) and cultivated overnight at 37 °C and 5% CO_2_ in Dulbeccos’s Modified Eagle Medium (Sigma Aldrich) containing L-Glutamin (200 mM, Sigma Aldrich), Amphotericin B (25 µg/mL, Sigma Aldrich), fetal calf serum (FCS, 10%, Sigma Aldrich) penicillin, and streptomycin (10 U/mL, Sigma Aldrich). After overnight culture, ME-CSCs and ACSCs were treated with LPS (LPS, 100 ng/mL, rough strain from *Salmonella enterica* Re 595, Sigma Aldrich), TNFα (10 ng/mL, PeproTech) or heat killed bacteria (HBK, 10^8^ cells/mL of heat killed *Staphylococcus aureus,* InvivoGen, Toulouse, France). Commercially purchased LPS from *Rhodobacter sphaeroides* (LPS-RS, 10,000 ng/mL, InvivoGen) served as TLR4 antagonist and was applied simultaneously to LPS from *Salmonella enterica* Re 595 (100 ng/mL, Sigma Aldrich). Controls were treated within medium described above without additional stimuli. For gene expression analysis, treatments were performed for 4 h, while a 2 h treatment was done for immunocytochemistry in accordance to our previous studies [22], and a 5 h treatment for western blot.

### 2.5. qPCR

RNA isolation was done with the innuPREP RNA mini Kit (Analytik Jena, Jena, Germany) and cDNA was synthesized using RevertAid First Strand cDNA Synthesis Kit (Thermo Fisher Scientific) according to the manufacturer’s guidelines. qPCR was performed as technical triplicates using the Luna^®^ Universal qPCR Master Mix (BioLabs, Frankfurt am Main, Germany) according to manufacturer’s guidelines in the MIC qPCR cycler (Bio Molecular Systems, San Francisco, USA). GAPDH served for normalization of cycle threshold values. GraphPad Prism Software (GraphPad Software, La Jolla, CA, USA) was used for statistical analysis. Expression levels were normalized to 100% for each target gene and donor. Primer sequences are depicted in Table 1.

### 2.6. Immunocytochemistry

ME-CSCs and ACSCs, treated as described above, were fixed with 4% PFA (Sigma Aldrich) for 20 min followed by permeabilization and blocking in TritonX-100 (AppliChem, Darmstadt, Germany) with 5% goat serum for 30 min. Primary antibody anti-NF-κB p65 (F-6) (sc-8008, Santa Cruz Biotechnologies, Heidelberg, Germany) was applied for 3 h at RT or overnight at 4 °C, while anti-phospho-c-Jun (S63) antibody (P05412, R&D Systems, Minneapolis, MN, United States) was applied for 3 h at RT. Secondary fluorochrome-conjugated antibodies anti-rabbit Alexa 488 and anti-mouse Alexa 555 (A32731, 21422, Molecular Probes, Thermo Fisher Scientific) were subsequently applied for 1 h at RT. Finally, nuclear counter staining with 4′,6-Diamidin-2-phenylindol (DAPI, 1:2000, Sigma Aldrich) for 15 min at RT was followed by mounting with Mowiol. Imaging was done using confocal laser scanning microscopy (LSM 780, Carl Zeiss, Jena, Germany) with ZEN software. For analysis of images, Fiji [23] was applied and the data was processed by GraphPad Prism Software (GraphPad Software).

### 2.7. Western Blot

ME-CSCs and ACSCs were stimulated as described above followed by harvesting via trypsination and subsequent lysis with lysis buffer (0.01 M Tris, 3 mM EDTA, 1% SDS) for 10 min at RT and denaturation for 10 min at 95 °C. The Protein quantification was performed using of Roti-Quant (Carl Roth, Karlsruhe, Germany) following manufacturer’s guidelines. Samples were mixed with 4× loading dye, incubated for 5 min at 95 °C and subjected to electrophoresis on 10% denaturing SDS polyacrylamide gels and subsequently transferred to a nitrocellulose membrane using the Biometra Fastblot B34 blotter (Analytik Jena AG, Jena, Germany). Blocking was subsequently performed with 10% milk powder and 0.1% Tween-20 in PBS for 30 min followed by application of the first antibody against A20 (A-12, sc-166692, Santa Cruz Biotechnologies) and overnight incubation at 4 °C. Afterwards, HRP-linked secondary antibody (515-035-003, Dianova, Hamburg, Germany) was applied for 1.5 h at RT. Visualization was performed via enhanced chemiluminescence using Fusion Solo (PEQLAB, Erlangen, Germany), while GAPDH (sc-32233, Santa Cruz Biotechnologies) served as loading control.

## 3. Results

### 3.1. Successful Isolation of Sphere-Forming Stem Cells from Middle Ear Cholesteatoma Tissue and Auditory Canal Skin

For isolation of stem cells, cholesteatoma tissue was obtained from the human posterior epitympanon, while auditory canal skin originated from the tympanomeatal flap (Figure 1A). Histological examination of the isolated cholesteatoma tissue revealed the presence of characteristic cholesteatoma structures like the perimatrix, the matrix, and enclosed cystic content (Figure 1B–C). On the contrary, a characteristic epithelial and a basal layer, as well as the dermis, were observable in sectioned auditory canal skin (Figure 1B–C). In accordance to our previous findings [20], we successfully isolated stem cells from the matrix and perimatrix of middle ear cholesteatoma (middle ear cholesteatoma stem cells, ME-CSCs) and from the dermis of auditory canal skin (auditory canal skin stem cells, ACSCs). Cultivated ME-CSCs and ACSCs revealed the ability of sphere formation under serum-free culture conditions (Figure 1D).

### 3.2. Stem Cells Derived from Middle Ear Cholesteatoma Show Significantly Elevated Expression of TRL4 Compared to Auditory Canal Skin Stem Cells

To determine a potential contribution of ME-CSCs to the pro-inflammatory environment of cholesteatoma, we investigated expression levels of TLR4 in ME-CSCs and ACSCs. Notably, a significantly elevated expression of TLR4 was observable in ME-CSCs from three independent donors in comparison to ACSCs derived from the corresponding donors (Figure 2A,B). On the contrary, transcript levels of TLR2 were not significantly elevated in ME-CSCs compared to ACSCs (Appendix A).

### 3.3. Treatment of ME-CSCs with LPS Results in Significantly Enhanced Expression Levels of Pro-Inflammatory Mediators Compared to ACSCs

Since bacterial infections are strongly associated to cholesteatoma growth and progression [24], we found TLR4 to be highly upregulated in ME-CSCs. ME-CSCs and ACSCs were exposed to the TLR4 agonist LPS from *S. enterica* in order to model the cholesteatoma microenvironment. Treatment of ME-CSCs from three independent donors with LPS resulted in a strong and highly significant upregulation of pro-inflammatory mediators IL- 1α, IL-1ß, IL-6, and IL-8 in comparison to untreated control (Figure 2C and Appendix A). LPS-treated ACSCs likewise revealed a slight increase in the expression of IL- 1α, IL-1ß, IL-6, and IL-8 compared to untreated control. However, the expression levels of these pro-inflammatory genes were significantly elevated in LPS-treated ME-CSCs in comparison to LPS-treated ACSCs for all three independent donors (Appendix A). Notably, we also observed a strong upregulation of TNFα expression in ME-CSCs induced by LPS-treatment in comparison to untreated controls and particularly to LPS-treated ACSCs, which showed no significant increase in the expression of TNFα after exposure to LPS (Figure 2C and Appendix A). Interestingly, expression levels of IL-18 were not affected by LPS-treatment in ME-CSCs or ACSCs, although ACSCs showed elevated basal expression levels in comparison to ME-CSCs (Figure 2C). To exclude a potential impact of TLR2-dependent signaling in ME-CSCs, we exposed ME-CSCs and ACSCs to heat killed bacteria (*S. aureus*, HKB). In addition to the unchanged expression levels of TRL2 between ME-CSCs and ACSCs, neither of the stem cell populations showed an upregulation of IL1-ß or IL-8 through stimulation with HBK (Appendix A). Hence, ME-CSCs are highly subjected to TLR4-dependent pro-inflammatory signaling, which is much more pronounced in ME-CSCs in relation to ACSCs. 

### 3.4. LPS-Dependent Pro-Inflammatory Gene Expression in ME-CSCs Is Mediated by an Enhanced Activity of NF-κB 

To determine potential downstream mediators of TLR4-dependent pro-inflammatory signaling in ME-CSCs, we investigated the effects of LPS on the activity of NF-κB p65 in ME-CSCs and ACSCs by immunocytochemistry. Upon treatment with LPS, ME-CSCs exhibited a distinct nuclear translocation of NF-κB p65 protein compared to control (Figure 3A, left panels, arrows). As a profound difference to ME-CSCs, only a minor nuclear translocation of NF-κB p65 was observable in ACSCs even after exposure to LPS (Figure 3A, right panels, arrowheads). An excessive quantification of the relative NF-κB p65 nuclear to cytoplasmic fluorescence intensities validated these observations in three independent donors. It revealed a significantly increased nuclear localization of NF-κB p65 in ME-CSCs when compared to control and LPS-treated ACSCs (Figure 3B). Notably, a slight basal translocation of NF-κB p65 could be detected in ME-CSCs without LPS-dependent stimulation (Figure 3A,B).

In addition, two out of three ACSC-populations derived from independent donors showed no significant changes in the amount of nuclear NF-κB p65 protein even after LPS-treatment (Figure 3B). Interestingly, we also observed a significant translocation of AP1 protein upon LPS-treatment in ME-CSCs compared to control and LPS-stimulated ACSCs, which displayed no increase in nuclear AP1 protein by LPS (Appendix A).

In accordance to the elevated nuclear translocation of NF-κB p65, expression levels of the specific NF-κB target genes A20 and IκBα were significantly increased in ME-CSCs upon LPS-stimulation compared to control and LPS-treated ACSCs (Figure 4A,B, Appendix A). Western Blot analysis further confirmed the elevated expression of A20 in LPS-treated ME-CSCs on protein level. In accordance to the transcriptional data, ME-CSCs exposed to LPS revealed a highly increased A20 protein amount compared to control and LPS-treated ACSCs. However, LPS-treated ACSCs showed no increase in A20 protein in comparison to control ACSCs (Figure 4C). These findings demonstrate NF-κB as the mediator of LPS-dependent pro-inflammatory gene expression exclusively in ME-CSCs and not in ACSCs.

### 3.5. ME-CSCs Show a TNFα-Mediated Feed-Forward-Loop of Pro-Inflammatory NF-κB Target Gene Expression

With TNFα being a major target gene of NF-κB and also an important cytokine involved in cholesteatoma progression [8], we investigated a potential pro-inflammatory feed-forward-loop of NF-κB target gene expression in ME-CSCs mediated by TNFα. We found that the expression of the TNF receptors TNFR1 and TNFR2 did not differ significantly between ME-CSCs and ACSCs (Figure 5A). But besides that, stimulation of ME-CSCs with TNFα resulted in significantly increased expression levels of IL-6, IL-8, and A20, and TNFα itself in comparison to control and TNFα-treated ACSCs (Figure 5B and Appendix A). Interestingly, we observed no significant difference in the expression levels of IL-1ß and IκBα between ME-CSCs and ACSCs after TNFα-dependent stimulation, although the expression levels were significantly elevated compared to controls (Figure 5B). In comparison to untreated control, exposure to TNFα further resulted in a highly pronounced and significant increase of TNFα expression almost exclusively in ME-CSCs and not in ACSCs (Figure 5B and Appendix A). The bottom line is that TNFα-induced expression of TNFα was significantly increased in ME-CSCs compared to ACSCs (Figure 5B and Appendix A). These observations indicate the presence of an exaggerated pro-inflammatory TNFα-mediated feed-forward-loop of NF-κB target gene expression in ME-CSCs.

### 3.6. Functional Inactivation of TLR4 via a TLR4-Antagonist Blocks Pro-Inflammatory Signaling in ME-CSCs

To functionally block TLR4-dependent pro-inflammatory signaling in ME-CSCs, we applied the TLR4 antagonist LPS from Rhodobacter sphaeroides (LPS-RS), which is structurally similar to the drug Eritoran [25] (Figure 6A). In accordance to our observation in ME-CSCs from donors I-III, we observed a strongly and significantly increased expression of IL-ß, IL-6, IL-8, TNFα, and A20 in ME-CSCs derived from donor IV after treatment with LPS from *S. enterica* compared to control and LPS-treated ACSCs (Figure 6B–F). Application of the TLR4 antagonist LPS-RS in ME-CSCs simultaneously treated with LPS from *S. enterica* resulted in a strong reduction of TNFα expression compared to LPS-treated ME-CSCs (Figure 6E). Although detected expression levels were overall lower compared to ME-CSCs, ACSCs treated with LPS and LPS-RS likewise showed a significantly impaired expression of IL-1ß, IL-6, IL-8, TNFα, and A20 compared to exposure to LPS. Notably, we observed an significant downregulation of expression of IL-1ß, IL-6, IL-8, and A20 in ME-CSCs exposed to LPS and the TRL4 antagonist LPS-RS compared to LPS-treated and especially untreated ME-CSCs (Figure 6B,C,E). 

## 4. Discussion

In the present study, we demonstrate that stem cells residing in cholesteatoma, a severe expanding lesion in the middle ear, mediating its inflammatory environment in a TLR4-NF-κB-dependent manner. 

We took advantage of previously described cholesteatoma stem cells as a cellular in vitro model of cholesteatoma inflammation [20]. In particular, we previously showed that Integrin-β1 ME-CSCs are present in the matrix and perimatrix of middle ear cholesteatoma. These ME-CSCs were able to differentiate into keratinocyte-like cells after exposure to growth factors like hepatocyte growth factor and keratinocyte growth factor present in cholesteatoma tissue [20]. In line with their expression of markers characteristic for neural-crest-derived stem cells (NCSCs) [26] and the commonly known contribution of NCSCs to middle ear development [20], ME-CSCs are proposed to be epidermal stem cells of neural crest origin. Accordingly, ME-CSCs were able to grow as spheres under serum-free conditions in previous studies [20] and the present study, thus showing a characteristic of epidermal stem cells (reviewed in [27]). With regard to the identification of epidermal stem cells in the tympanic membrane [28], ME-CSCs showed markers of epidermal stem cells. Hence, they are particularly suggested to migrate from this stem cell niche to the lesion and contribute to cholesteatoma through differentiation into proliferating keratinocyte-like cells [20]. Here, we extend these promising findings by demonstrating a role of ME-CSCs in cholesteatoma formation as mediators of its inflammatory environment. In particular, ME-CSCs exhibited a pronounced inflammatory footprint characterized by a significantly increased expression of the TLR4, a receptor recognizing bacterial LPS in the first line defense against bacterial infections. An enhanced expression of TLR4 was already shown in 2013 by Hirai and colleagues in middle ear tissues obtained from five patients with acquired middle ear cholesteatoma [13]. In 2015, Si and colleagues extended these findings by demonstrating TLR4 as a major driver of cholesteatoma pathogenesis in terms of promoting both inflammation and bone destruction. Expression of TLR4 was particularly shown to be correlated with disease severity in terms of increased invasion, bone destruction, and hearing loss [14]. Although an up-regulation of TLR2 was observable in cholesteatoma tissue [13], deficiency in TLR2 in mice did not affect disease severity or inflammatory responses [14]. Interestingly, we observed an upregulation of TLR4 gene expression in ME-CSCs compared to ACSCs in the present study, whereas expression of TLR2 and TLR2-dependent inflammatory signaling remained unaffected in ME-CSCs.

Aggressiveness of cholesteatoma is strongly associated with the presence of bacteria [24] and bacterial LPS [10] as the common agonist of TLR4. LPS was also demonstrated to directly trigger pathogenic characteristic of cholesteatoma [29,30]. Hence, we utilized LPS to investigate pro-inflammatory signaling in ME-CSCs. LPS is well-described to initiate pro-inflammatory signaling by binding to a heterodimer consisting of TLR-4 and myeloid differentiation factor 2 (MD-2), in turn leading to recruitment of MyoD88 and phosphorylation of the IκB kinases (IKKs). Phosphorylated IKKs in turn phosphorylate IκBα, resulting in its polyubiquitylation and 26S-proteasome-mediated degradation. Loss of IκBα leads to unmasking of the nuclear translocation signal region of the NF-κB p65/p50 heterodimer, resulting in its nuclear translocation and activation of distinct target genes via binding to κB-sites [31,32]. NF-κB target genes particularly include pro-inflammatory cytokines like TNFα, IL-1α, IL-1ß, IL-6, and IL-8 [33,34,35,36] and were shown to be upregulated in cholesteatoma tissue [8,9,12]. Consequently, we demonstrate that exposure of ME-CSCs to LPS resulted in significantly increased nuclear translocation of NF-κB p65 resulting in a strongly enhanced pro-inflammatory gene expression pattern of TNFα, IL-1α, IL-1ß, IL-6, and IL-8 compared to ACSCs. When compared to ACSCs, ME-CSCs further demonstrated a strongly elevated expression of the NF-κB target genes IκBα and A20 if exposed to LPS. In this line, A20 was already described to be higher expressed in cholesteatoma tissue using whole human genome microarrays [37], which we could verify in case of the A20 expression in ME-CSCs on protein level. In addition, we observed an increase in nuclear translocation of the transcription factor AP1, suggesting a pro-inflammatory signaling mediated by both NF-κB and AP1 in ME-CSCs. Being a dimer of c-Jun and c-Fos, AP-1, as well as NF-κB, are both well-described to drive transcription of cytokines like TNFα and IL-1β in inflammatory diseases [38]. 

As a major target gene of NF-κB, TNFα itself is known to drive a pro-inflammatory feed-forward-loop described in [39] by strongly activating the NF-κB transcriptional pathway [40]. Our present findings showed that stimulation of ME-CSCs with TNFα induces a distinctively elevated expression levels of IL-6, IL-8, A20, and TNFα in comparison to TNFα-treated ACSCs. This observation was very similar to the effect observed after stimulation with LPS. But in contrast to the TLR4, we could not observe a higher expression of the corresponding TNF receptors in ME-CSC. Hence, we suggest that the overlapping parts between the TNFα and LPS pathway [40] might be more sensitive in ME-CSC. Anyhow, the resulting TNFα-driven pro-inflammatory feed-forward-loop of NF-κB activation in ME-CSCs is in line with the already described involvement of TNFα in cholesteatoma progression [8] and other pathogenesis via inflammatory cues [41]. Additionally, the exaggerated inflammation might give rise to an enhanced epidermal differentiation observed in other inflammatory diseases like eczema [42], psoriasis [43], or nasal polyps [18]. Consequently, this might trigger the epidermal stem cells to differentiate into keratinized squamous epithelium promoting cholesteatoma formation.

As a clinical perspective, we applied the TLR4 antagonist LPS of *Rhodobacter sphaeroides* (LPS-RS)*,* which is structurally similar to the drug Eritoran, a synthetic molecule derived from the lipid A structure of LPS-RS [25]. In comparison to LPS-RS, the lipid A of Eritoran shows one hydrocarbon chain combining structural components of two out of five LPS-RS chains, while two further chains are nearly completely similar and one chain reveals complete structural similarity [44]. Eritoran was shown to bind competitively to the hydrophobic pocket of MD-2, thus preventing dimerization of TLR4-MD2 complexes (reviewed in [44]) and in turn intracellular downstream activation of NF-κB and activation of pro-inflammatory gene expression [25,45,46]. Even though there are other inhibitors of TLR4 signaling like Sulforaphane [47] we decided to focus on Eritoran in this study. Firstly, because it lays most upstream of the initial problem, the contamination of cholesteatoma tissue with LPS [11]. Secondly, because Eritoran was already successfully applied for treating chronic airway response to inhaled lipopolysaccharide in mice [48] and acute severe liver injury in rats [49]. Most importantly, Eritoran was applied in different clinical trials against sepsis e.g., phase II [50] and even phase III [51], reviewed in [41], therefore making it a promising molecule for new clinical application sometime soon. In the present study, we observed a pronounced reduction of the expression of IL-8 and A20 and a near-complete loss of expression of IL-1ß, IL-6, and TNFα in ME-CSCs exposed to both LPS and LPS-RS compared to LPS-treated and untreated ME-CSCs. These findings suggest a local treatment strategy for cholesteatoma by specifically targeting TLR4-mediated pro-inflammatory down-stream signaling in cholesteatoma stem cells as mediators of inflammation and cholesteatoma progression (Figure 7). The local drug administration after Cholesteatom surgery to the middle ear cavity might reduce recurrence of cholesteatoma. This could be realized by applying carrier substances e.g., chitosan, a biodegradable polymer, which is even applicable on the delicate inner ear [52].

Importantly, TLR4-deficiency in mice was already shown to be protective against cholesteatoma-driven hearing loss and bone destruction by reduction of local expression of pro-inflammatory cytokines and osteoclast formation [14]. In summary, we demonstrated that stem cells residing in cholesteatoma are the cellular mediators of its highly pro-inflammatory environment, suggesting a distinct role of ME-CSCs as drivers in cholesteatoma progression. On a molecular level, LPS-dependent pro-inflammatory gene expression patterns observed in ME-CSCs were driven by a TLR4-NF-κB signaling cascade, resulting in a TNFα-mediated feed-forward-loop of pro-inflammatory NF-κB target gene expression (Figure 7). Functional inactivation of TLR4 via the TLR4-antagonist LPS-RS blocked pro-inflammatory signaling in ME-CSCs (Figure 7), thus providing a direct clinical perspective for pharmaceutical cholesteatoma treatment that is not present today. 

## Figures and Tables

**Figure 1 cells-09-00199-f001:**
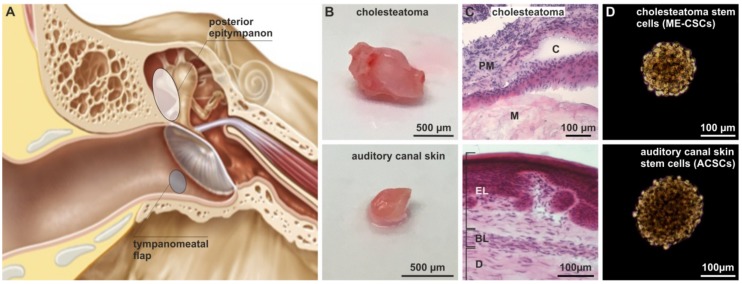
Isolation of middle ear cholesteatoma and auditory canal skin stem cells. (**A**) Schematic view on the localization of cholesteatoma obtained from the posterior epitympanon and the auditory canal skin removed from the tympanomeatal flap. Modified from [20] (Creative Commons Attribution 4.0 International License). (**B**) Obtained cholesteatoma and auditory canal skin tissue. (**C**) H&E staining of cryostat sections of cholesteatoma tissue revealed characteristic structures as the matrix (M), perimatrix (PM) and cystic content, while auditory canal skin revealed an epithelial layer (EL), a basal layer (BL), and a dermis (D). (**D**) Successfully isolated stem cells from cholesteatoma tissue and auditory canal skin were able to form spheres.

**Figure 2 cells-09-00199-f002:**
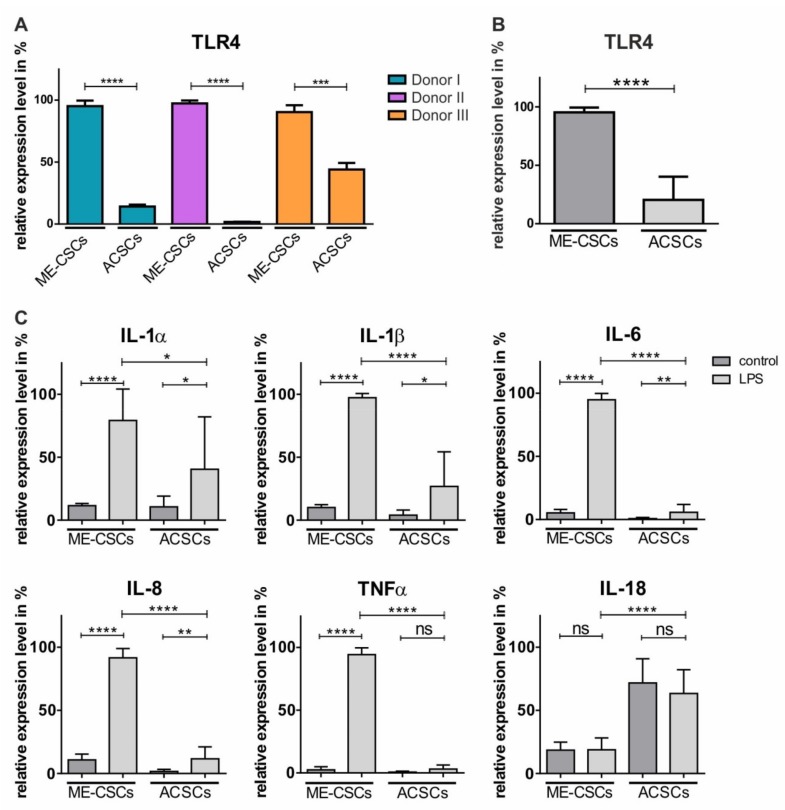
Compared to ACSCs, ME-CSCs show significantly increased expression of TRL4 accompanied by significantly enhanced LPS-dependent pro-inflammatory gene expression patterns. (**A**) qPCR analysis revealed highly increased expression levels of TLR4 in ME-CSCs compared to ACSCs in three different donors (**** ≤0.0001, *** ≤0.001, unpaired *t*-test, one-tailed, confidence interval: 95%) (**B**) Mean of the relative expression levels of TLR4 from ME-CSCs and ACSCs validated the increased expression levels of TLR4 in ME-CSCs compared to ACSCs (n = 3, **** ≤0.0001, Mann Whitney test, one-tailed, confidence interval: 95%). (**C**) Expression levels of the pro-inflammatory mediators IL-1α, IL-1ß, IL-6, IL-8, and TNFα were strongly increased in ME-CSCs treated with LPS from *S. enterica* compared to untreated control and LPS-treated ACSC. Expression level of IL-18 was not affected by LPS-treatment (mean of the relative expression levels from ME-CSCs and ACSCs (n = 3, **** ≤0.0001, ** ≤0.01, * ≤0.05, ns > 0.05, Mann Whitney test, one-tailed, confidence interval: 95%).

**Figure 3 cells-09-00199-f003:**
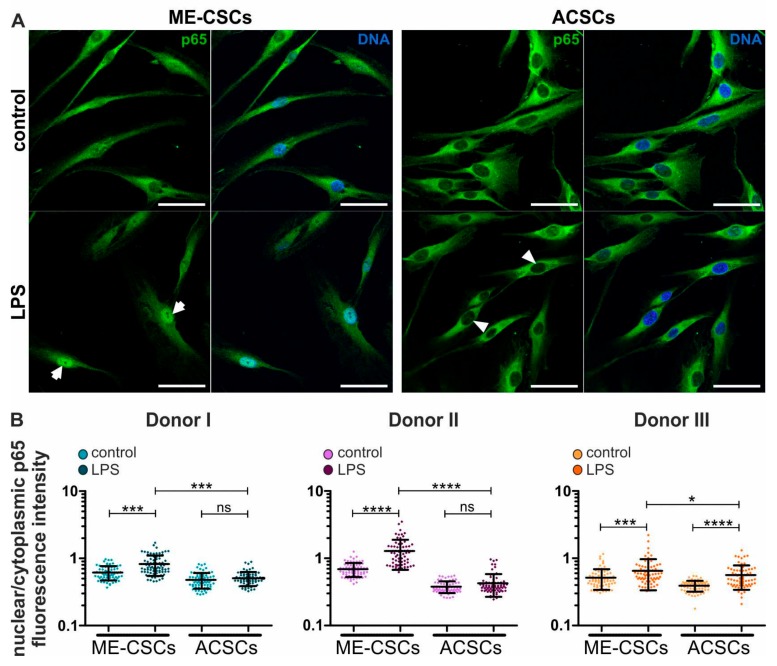
LPS-dependent pro-inflammatory gene expression in ME-CSCs is mediated by NF-κB. (**A**) Exemplary immunocytochemical staining reveals strong nuclear translocation of NF-κB p65 protein in LPS-treated ME-CSCs (arrows) compared to control ME-CSCs, which was only slightly detectable in both control and LPS-treated ACSCs (arrowheads). ME-CSCs and ACSCs were derived from Donor II, LPS from *S. enterica*, scale bars: 50 µm. (**B**) Quantification of NF-κB p65 nuclear to cytoplasmic fluorescence intensities validated a significantly increased nuclear localization of NF-κB p65 in ME-CSCs compared to control and to LPS-treated ACSCs (n = 3, **** ≤0.0001, *** ≤0.001, * ≤0.05, ns > 0.05, Mann Whitney test, one-tailed, confidence interval: 95%).

**Figure 4 cells-09-00199-f004:**
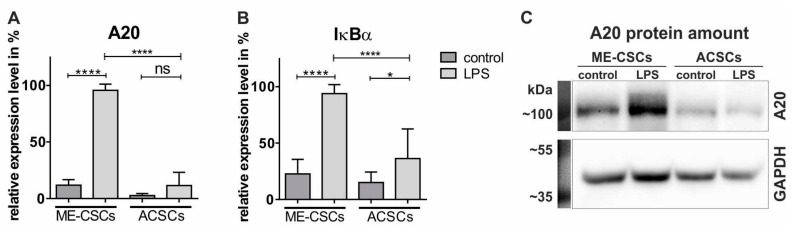
LPS-treatment of ME-CSCs results in significantly elevated levels of NF-κB target genes compared to ACSCs. (**A**–**B**) qPCR analysis showed significantly increased transcript levels of the NF-κB related targets A20 and IκBα in ME-CSCs after LPS-treatment (*S. enterica*) compared to control and LPS-stimulated ACSCs (mean of the relative expression levels from ME-CSCs and ACSCs, n = 3, **** ≤0.0001, * ≤0.05, ns > 0.05, Mann Whitney test, one-tailed, confidence interval: 95%). (**C**) Western blot analysis demonstrated highly increased amounts of A20 protein in LPS-treated ME-CSCs in comparison to ACSCs exposed to LPS. ME-CSCs and ACSCs were derived from donor IV.

**Figure 5 cells-09-00199-f005:**
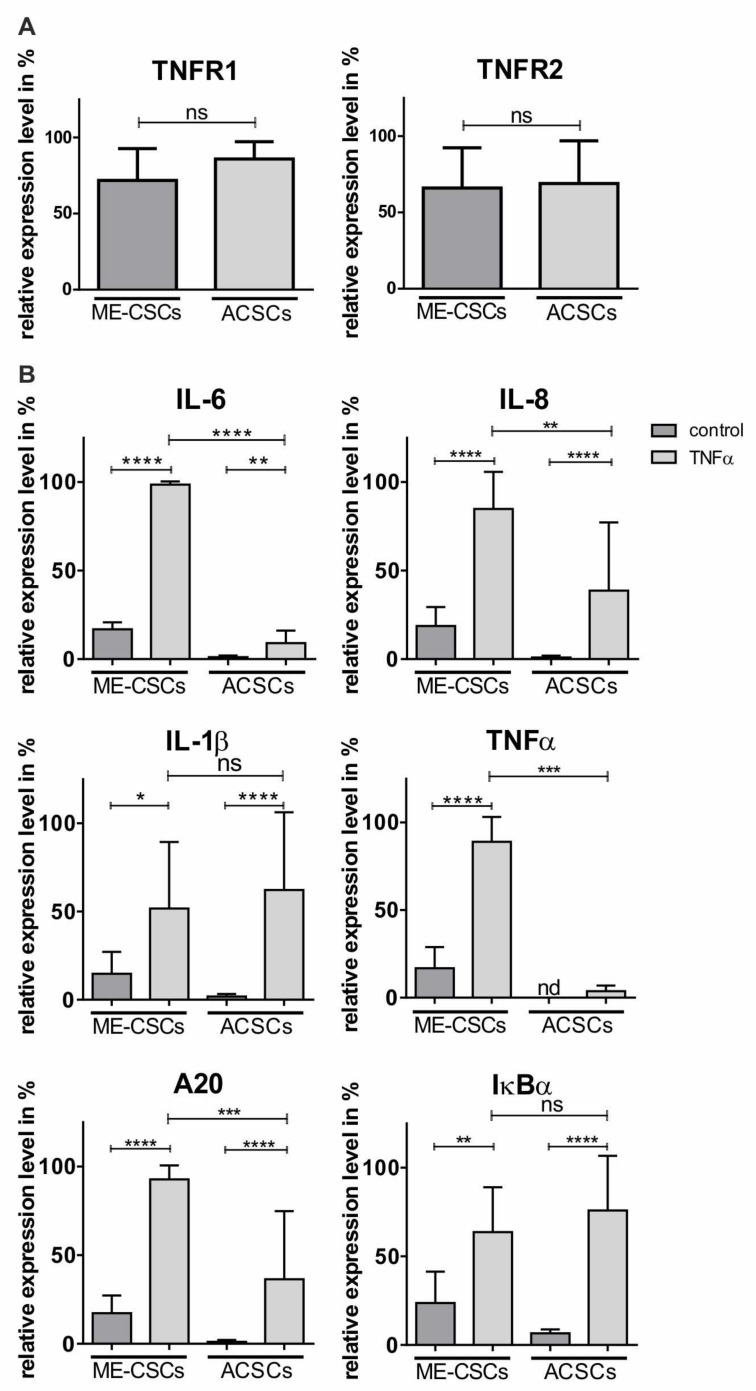
ME-CSCs reveal a TNFα-mediated feed-forward-loop of pro-inflammatory NF-κB target gene expression. (**A**) qPCR analysis revealed no significant difference in expression levels of TNFR1 and TNFR2 (mean of the relative expression levels from ME-CSCs and ACSCs, n = 3, ns > 0.05; Mann Whitney test, one-tailed, confidence interval: 95%). (**B**) qPCR analysis showed a significant increase in the expression levels of IL-6, IL-8, TNFα, and A20 in ME-CSCs stimulated with TNFα compared to control and TNFα-stimulated ACSC (mean of the relative expression levels from ME-CSCs and ACSCs, n = 3, **** ≤0.0001, *** ≤0.001, ** ≤0.01, * ≤0.05, ns > 0.05; Mann Whitney test, one-tailed, confidence interval: 95%, nd: not detectable).

**Figure 6 cells-09-00199-f006:**
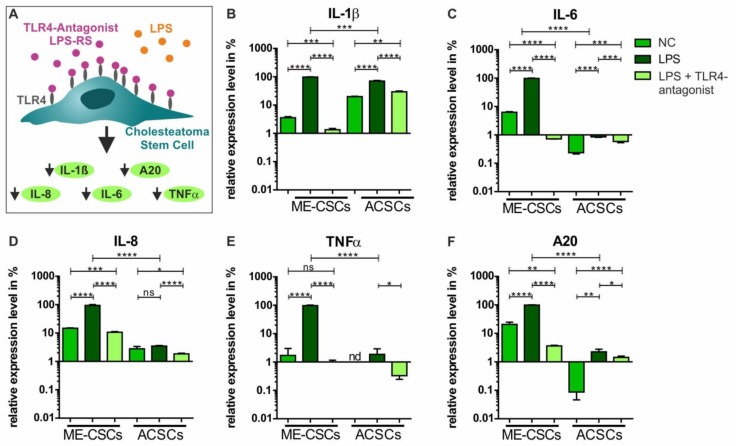
Functional inactivation of TLR4 via the TLR4-antagonist LPS-RS blocks pro-inflammatory signaling in ME-CSCs. (**A**) Schematic view on the mode of action of the TLR4 antagonist LPS-RS in ME-CSCs. (**B**–**F**) ME-CSCs simultaneously treated with the TLR4 antagonist LPS from *R. sphaeroides* (LPS-RS) and LPS from *S. enterica* (LPS) showed a significant reduction of IL-ß, IL-6 IL-8, TNFα, and A20 expression compared to LPS-treated ME-CSCs. Although detected expression levels were overall lower compared to ME-CSCs, ACSCs treated with LPS and LPS-RS likewise showed a significantly impaired expression of IL-ß, IL-6, IL-8, TNFα, and A20 compared to stimulation with LPS (ME-CSCs and ACSCs were derived from donor IV, **** ≤0.0001, *** ≤0.001, ** ≤0.01, * ≤0.05, ns > 0.05, unpaired *t*-test, one-tailed, confidence interval: 95%, nd: not detectable).

**Figure 7 cells-09-00199-f007:**
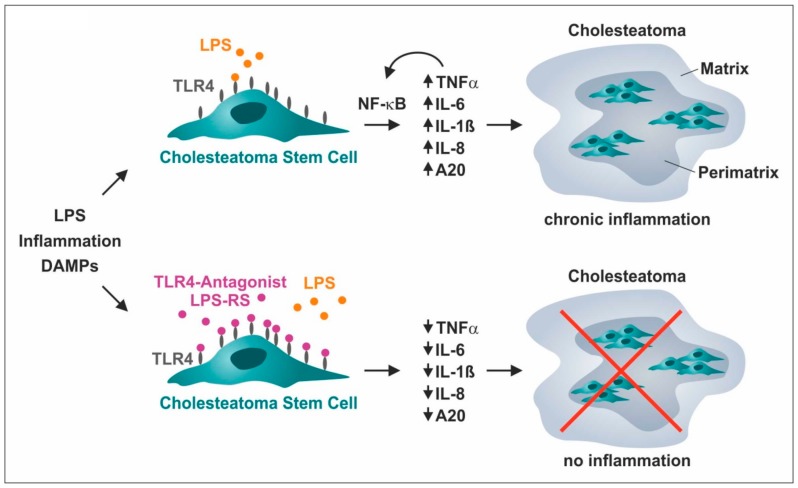
Schematic summary of cholesteatoma stem cells as cellular mediators of the highly pro-inflammatory environment in cholesteatoma and drivers of cholesteatoma progression. LPS-dependent pro-inflammatory gene expression in cholesteatoma stem cells was driven by a TLR4-NF-κB signaling cascade resulting in a TNFα-mediated autocrine feed-forward-loop of pro-inflammatory gene expression. As a clinical perspective, functional inactivation of TLR4 via the TLR4-antagonist LPS-RS blocked pro-inflammatory signaling, suggesting its therapeutic applicability for treating cholesteatoma.

**Table 1 cells-09-00199-t001:** Primer sequences.

Primer	Sequence (5′ to 3′)	Size of Product (bp)
**A20**	TACCCTTGGTGACCCTGAAG CCTTGGACGGGGATTTCTAT	175
**GAPDH**	CTGCACCACCAACTGCTTAG GTCTTCTGGGTGGCAGTGAT	108
**IL-18**	GCAAGGATTGTCTCCCAGT CGATCTGGAAGGTCTGAGGT	125
**IL-1α**	TGCCTGAGATACCCAAACC GCCAAGCACACCCAGTAGTC	145
**IL-1β**	TGTACCTGTCCTGCGTGTTGAAAG CTGGGCAGACTCAAATTCCAGCTT	149
**IL-6**	GCAAAGAGGCACTGGCAGAAAACA TTCTGCAGGAACTGGATCAGGACT	226
**IL-8**	TCTCTTGGCAGCCTTCCTGATTTC AGTTTTCCTTGGGGTCCAGACAGA	227
**IκBα**	AGACCTGGCCTTCCTCAACT GTCTCGGAGCTCAGGATCAC	127
**TLR2**	AGATGCCTCCCTCTTACCCATGTT AAGACTTTGGCCAGTGCTTGCT	186
**TLR4**	CACAGACTTGCGGGTTCTACATCA TGGACTTCTAAACCAGCCAGACCT	192
**TNFR1**	AGGGGACAGGGAGAAGAGAGGTT TTCTGAAGCGGTGAAGG	181
**TNFR2**	TCACCTCCAGCTCCACCTAT AGGCTCTGTGGCTTGTGG	175
**TNFα**	AAGCCCTGGTATGAGCCCATCTAT AGGGCAATGATCCCAAAGTAGACC	137

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
