# Peer review of "Stem Cell-Induced Inflammation in Cholesteatoma Is Inhibited by the TLR4 Antagonist LPS-RS"

_cells, 2020, doi:10.3390/cells9010199_

Round 1

Reviewer 1 Report

Thank you for inviting me to evaluate the article titled “tem cell-induced inflammation in cholesteatoma is inhibited by the TLR4 antagonist LPS-RS”. The subject addressed in this manuscript is worthy of investigation and the goals and the results of the study are clear. But there are still some questions and mistakes in the manuscript.  It is noted that the manuscript needs careful editing by someone with expertise in technical English editing paying particular attention to English grammar, spelling, and sentence structure so that the goals and results of the study are clear to the reader. the manuscript could be accepted after major revision. Anyway, some points are listed below for the author to improve the manuscript.

Line 14, keratinizing should be keratinized?

Line 21, TRL4 should be TLR4?

Line 26, IL-β should be IL-1β?

Line 109, Separation process of LPS-RS need write in the manuscript.

Line 112, “For gene expression analysis, treatments were performed for 4h, while a 2h treatment was done for immunocytochemistry and a 5h treatment for western blot.”  How do you choose these time nodes for experiments? How about other times?

Line 123, three wire table is more suitable.

Line 171, what is the protein level of TLR4 in Stem cells?

Line 212, How about the phosphorylation of NF-kB, and how long does the nuclear localization of NF-kB p65?

Line 284,need to give the structure of LPS-RS and Eritoran, and need to indicate similar groups.

Line 306, mediate should be mediating.

Author Response

It is noted that the manuscript needs careful editing by someone with expertise in technical English editing paying particular attention to English grammar, spelling, and sentence structure 

We now carefully revised the manuscript in terms of language.

Line 14, keratinizing should be keratinized?

We now changed the word accordingly.

Line 21, TRL4 should be TLR4?

We now changed the word accordingly.

Line 26, IL-β should be IL-1β?

We now changed the word accordingly.

Line 109, Separation process of LPS-RS need write in the manuscript.

We thank the referee for this remark. As we now stated in the manuscript (lines 129-130), LPS-RS was commercially purchased and thus required no further separation processes prior to application.

Line 112, “For gene expression analysis, treatments were performed for 4h, while a 2h treatment was done for immunocytochemistry and a 5h treatment for western blot.”  How do you choose these time nodes for experiments? How about other times?

Thank you for raising this issue. We choose the respective time points in accordance to our previous studies (Müller et al., 2016 Clin Sci (Lond), 130 (15), 1339-52 2016). Here, treatment of adult human nasal stem cells with LPS resulted in increased nuclear translocation of NF-kB p65 after two hours observed by immunocytochemistry. In line with the present findings, analysis of target gene expression was performed after four hours. For the western blot analysis, we choose a later time point (five hours)  to allow not only transcription, but also translation of NF-kB target gene mRNA. We therefor did not investigate further time points in the present study and included the aspects discussed above in the materials and methods section (see lines 129-130)

Line 123, three wire table is more suitable.

We thank the reviewer for this remark. Unfortunately, it is unclear to us how exactly the table has du be improved and in which regard this improvement facilitates better guidance for the reader. We would be glad, if the referee may clarify the specific changes to modify the table towards a “three wire table” or provide an example.

Line 171, what is the protein level of TLR4 in Stem cells?

We thank the referee for raising this interesting question. Although our present manuscript mainly focuses on the analysis of mRNA, it would be interesting to assess the level of TLR4 in ME-CSCs on protein level. Due to the only very short time for addressing the reviewers comments (10 days), we are not able to address this matter on an experimental level. However, we suggest the enhanced mRNA levels of TRL4 observed in the present study to build the basis of increased amounts of TLR4 on protein level, which is in line with a range of studies determining TRL4 expression in cholesteatoma (Hirai et al., 2013, International journal of pediatric otorhinolaryngology, 77, 674-676; Si et al. 2015, Scientific reports 5, 16683; Iino et al., 1990, Acta Otolaryngol, 110, 410-415; Kobayashi et al., 2005, Mediators Inflamm, 2005, 210-215).

Line 212, How about the phosphorylation of NF-kB, and how long does the nuclear localization of NF-kB p65?

Thank you for these questions. While phosphorylation of NF-kB was not determined in the present study, we observed nuclear translocation of NF-kB after 2 hours of treatment with LPS. Although TNFα-dependent nuclear translocation of NF-kB p65 was reported to last for at least 50 min  in HeLa cells, (JBC 273, 28897-28905), neural stem cells show nuclear NF-kB even after 30 Minutes of TNF-α stimulation (BMC Neuroscience volume 7, Article number: 64 (2006)). In microglia, LPS-dependent stimulation resulted in nuclear translocation of NF-kB after 1 hour (Gaikwad et al., Int J Inflam. 2015; 2015: 361326.) In line with our present findings,  treatment of adult human nasal stem cells with LPS resulted in increased nuclear translocation of NF-kB p65 after two hours (Müller et al., 2016 Clin Sci (Lond), 130 (15), 1339-52 2016). Although nuclear translocation may thus start at an earlier time point, our observations indicate a pronounced nuclear localization of NF-kB lasting for at least up to two hours. With regard to the observed nuclear translocation and the broad range of upregulated target genes on mRNA and protein level, we suggest our findings to sufficiently demonstrate the activation of NF-kB. We did thus not additionally focus on the phosphorylation of NF-kB, but included the ratio of choosing a 2 h time point for assessing its nuclear translocation (see above) in the materials and methods section (see line 129-130).

Line 284,need to give the structure of LPS-RS and Eritoran, and need to indicate similar groups.

We thank the referee for this remark. The structure of both LPS-RS and Eritoran is known to be structurally highly similar in terms of the hydrocarbon chains in the lipid A. While the lipid A of LPS-RS comprises 5 hydrocarbon chains, two of these are combined to one in Eritoran, which reveals 4 hydrocarbon chains. This combined hydrocarbon chains shows structural hallmarks (one carbonyl-group and a double bond) similar to the two respective initial chains in LPS-RS. Two other chains are nearly similar despite of few single hydroxyl- and carbonyl-groups and the last hydrocarbon chain of Eritoran and LPS-RS is completely similar (Barochia et al.,Expert Opin. Drug Metab. Toxicol. (2011) 7(4); (Rossignol DP, Lynn M. Curr Opin Investig Drugs 2005;6(5):496-502). Since the structure is commonly available, we only included this aspect in more detail in the respective part of the discussion sections (see line 407-409).

Line 306, mediate should be mediating.

We now changed the word accordingly.

Reviewer 2 Report

In this study, Schürmann and colleagues investigated the role of middle ear cholesteatoma stem cells (ME-CSCs) in chronic inflammation in cholesteatoma. They found ME-CSCs showed a significantly increased expression of TRL4 and pro-inflammatory gene expression compared to non-pathogenic control cells. Moreover, functional inactivation of TLR4 via the antagonist LPS-RS blocked chronic inflammation in ME-CSCs. Overall, the study is well-designed and presented. Following concerns should be addressed in revision.

Why were ACSCs used as control in the study? Is there any difference between healthy individual and cholesteatoma patient's ACSCs? The authors compared TLR4 and TLR2 expression between ME-CSCs and ACSCs. It would be more informative to measure a broader panel of pattern recognition receptors (other TLR, NLR, RLR, DNA sensors).   Does TLR agonist triggers differentiation of ME-CSCs or change their stemness? Why TLR4 is upregulated in ME-CSCs? Was it caused by inflammatory microenviroment in middle ear? Or it is the driving force of inflammation.

Author Response

Why were ACSCs used as control in the study? Is there any difference between healthy individual and cholesteatoma patient's ACSCs?

We thank the reviewer for raising these important questions. In a previous study (Scientific Reports, (2018) 8:6204), we demonstrated the isolation of ACSCs from healthy auditory canal skin samples of cholesteatoma patients. ACSCs and ME-CSCs shared similar epidermal stem cell characteristics like the abilities to form spheres, self-renew and undergo multilineage differentiation. Most theories describing cholesteatoma formation assume a disturbance of the tympanic membrane resulting in deposition of auditory canal tissue into the middle ear, which might also comprise the predecessor cells of ME-CSC. Notably, exposure of ME-CSCs to factors mimicking the microenvironment of the cholesteatoma (KGF, EGF, HGF and IGF-II) resulted in differentiation into keratinocyte-like cells, which was not observable in ACSCs (Scientific Reports, (2018) 8:6204). We therefore concluded that ACSCs are suitable control cells for ME-CSC, since they share certain epidermal stem cells characteristics, but lack the pathogenic phenotype of ME-CSCs. However, we were not able to assess any differences between ACSCs derived from cholesteatoma patients and healthy individuals since the biopsy auditory canal skin samples from healthy donors, especially that close to the tympanic membrane, cannot be performed for ethical reasons. In addition, the already high variability of cell behavior observed between ME-CSCs/ACSCs from donors may be even more pronounced when comparing donor-independent control cells and thus limit the usability of ACSCs from healthy individuals.

The authors compared TLR4 and TLR2 expression between ME-CSCs and ACSCs. It would be more informative to measure a broader panel of pattern recognition receptors (other TLR, NLR, RLR, DNA sensors).  

We thank the referee for this remark. On a microbiological level, cholesteatomas are mainly associated with biofilms composed of Gram-positive as well as Gram-negative bacteria and their respective endotoxins being recognized via TLR2 or TLR4. In the present study, we thus focused on the TLR4 and TLR2 and took advantage of the observation that particularly the TRL4 is known as one major driver of cholesteatoma progression (Hirai et al., 2013, International journal of pediatric otorhinolaryngology, 77, 674-676; Si et al. 2015, Scientific reports 5, 16683; Iino et al., 1990, Acta Otolaryngol, 110, 410-415; Kobayashi et al., 2005, Mediators Inflamm, 2005, 210-215). In addition, LPS was also was shown to be present in cholesteatoma (Peek et al., Otol Neurotol 2003, 24, 709-713)  and demonstrated to directly trigger pathogenic characteristic of cholesteatoma (Kobayashi et al., Mediators Inflamm 2005, 2005, 210-215, Iino et al., Acta Otolaryngol 1990, 110, 410-415). These findings are discussed in great details in the discussion section (lines 352-367) and emphasize our focus of investigating the expression of TLR4 and TLR2 rather than a broader panel of pattern recognition receptors. In addition, we are not able to address this matter on an experimental level due to the only very short time for addressing the reviewer’s comments (10 days). Since the assessing the potential expression levels of further pattern recognition receptors in ME-CSCs and ACSCs is nevertheless quite interesting, we will aim for addressing this matter in future studies. Particularly the receptors able to detect prokaryotic pathogens are interesting aims in this regard e.g. the NOD1 and NOD2 able to recognize different peptidoglycans.

Does TLR agonist triggers differentiation of ME-CSCs or change their stemness?

Thank you for this very interesting question. We are just starting to do experiments on exactly this issue with our in vitro model and looking forward to gain interesting results in the future. Although we are therefore not able to address this completely new topic in the present manuscript, we added a whole new paragraph focusing on this topic, which allows the reader to grasp this interesting idea:

“Anyhow, the resulting TNFα-driven pro-inflammatory feed-forward-loop of NF-kB activation in ME-CSCs, is in line with the already described involvement of TNFα in cholesteatoma progression [8] and other pathogenesis via inflammatory cues [41]. Additionally to this, the exaggerated inflammation might give rise to an enhanced epidermal differentiation observed in other inflammatory disease like eczema [42], psoriasis [43] or nasal polyps [20]. Consequently this might trigger the epidermal stem cells to differentiate into keratinized squamous epithelium promoting cholesteatoma formation.” (lines 397-404).

Accordingly, we also changed respective sections in the discussion to emphasize the epidermal origin of ME-CSCs (lines 336-345): “In particular, we previously showed that Integrin-b1 positive ME-CSCs are present in the matrix and perimatrix of middle ear cholesteatoma. […] The derived ME-CSCs were able to grow as spheres under serum-free conditions in previous [19] and the present study, thus showing a further characteristic of epidermal stem cells [reviewed in 28].”

Why TLR4 is upregulated in ME-CSCs? Was it caused by inflammatory microenvironment in middle ear? Or it is the driving force of inflammation. 

We thank the referee for raising this important issue. The expression of TLR4 is known to be upregulated in different diseases associated with chronic inflammation and particularly in cholesteatoma (Hirai et al., 2013, International journal of pediatric otorhinolaryngology, 77, 674-676; Si et al. 2015, Scientific reports 5, 16683; Iino et al., 1990, Acta Otolaryngol, 110, 410-415; Kobayashi et al., 2005, Mediators Inflamm, 2005, 210-215). In addition to the discussion of the role of TLRs in cholesteatoma and ME-CSCs (lines 350-362), we now included a further linkage between NF-kB activity and hyper-proliferative mucosal tissue in the introduction section (lines 64-65): “For example the hypertrophy of nasal mucosa was linked to enhanced activity of NF-kB [20] in chronic rhinosinusitis exhibiting overexpression of TLR4 [21].” Although we can only suggest if ME-CSCs are the cause or consequence of chronic inflammation, our present observations emphasize their distinct role in further mediating its highly pro-inflammatory environment, suggesting a distinct role of ME-CSCs as drivers in cholesteatoma progression. From our point of view, ME-CSCs thus represent one major driving force of inflammation, since they are more susceptible for inflammatory stimuli and may thus facilitate chronic inflammation. On the contrary, the initial inflammatory environment may cause the presence of ME-CSCs in the first place, although we can only speculate regarding this issue. In addition, ME-CSCs may contribute to cholesteatoma progression in terms of differentiation, which we are currently investigating and have now included in the discussion section (see above and lines 400-404). However, the present study determined ME-CSCs as regulators of the inflammatory environment of cholesteatoma via TLR4-mediated NF-kB-signaling with TLR4 as a therapeutic target.

Reviewer 3 Report

I am impressed by this paper for authors in this paper transmit several interesting and important information regarding the pathophysiology of  cholesteatoma, they suggested that the key factor TLR4, which is originated from ME-CSCs, dominates the leads the inflammation to the formation of the cholesteatoma and other situations. This paper is deserved for publication.

I only have some suggestions

abstract: Facing this challenge-----sounds strange to me. To rephrase the sentence may make the whole better. 

                    In summary ...... this sentence is very confusing. What is your therapeutic target? stem cells, or stem cells with TLR4, or TLR4? 

My suggestion is :

In summary, we demonstrate that ME-CSCs mediate the inflammatory environment of cholesteatoma via TLR4-mediated NF-kB-signaling. Targeting TLR4 may quench the formation and progression of cholesteatoma

Introduction: In summary, we were able to determine ME-CSCs as cellular mediators of the inflammatory environment of cholesteatoma via TLR4-mediated NF-kB-signaling.

This is another confusing sentence, because I thought your main idea is ME-CSCs, which were stimulated by inflammation, form and progress cholesteatoma. Can you make it more clear regarding your findings?

Results:

Figure 1: figures were good. However, it will be optional if you can add stem cell fluorescent pictures overlapping with common and different stem cell markers to differentiate ME-CSCs and ACSCs.

Figure 3. If translocation of NF-KB p65 is induced by LPS, you should show your readers that the use of LPS-RS can reverse the translocation. Personally, I am so curious the TNF-alpha may cause the translocation ?  If so, the figure 7 may be more intact and complete.

Have your ever tried TLR4 inhibitor Sulforaphane (SFN) [1-isothiocyanato-4-(methylsulfinyl)butane]? Would effects of SFN are similar or different to LPS-RS?

Discussion:  too many redundant and repeated words, like many “accordingly”….

Some syntax errors in the content need to be revised.

Author Response

I only have some suggestions

Abstract: Facing this challenge-----sounds strange to me. To rephrase the sentence may make the whole better. 

We now rephrased the sentence as follows: “In the presented study, we utilized our established human in vitro cholesteatoma stem cell model for treatments with LPS, TNFα and the TLR4-antagonist LPS-RS followed by qPCR, western blot and immunocytochemistry.”

In summary ...... this sentence is very confusing. What is your therapeutic target? stem cells, or stem cells with TLR4, or TLR4?  My suggestion is:  In summary, we demonstrate that ME-CSCs mediate the inflammatory environment of cholesteatoma via TLR4-mediated NF-kB-signaling. Targeting TLR4 may quench the formation and progression of cholesteatoma

Thank you for this remark, we suggest stem cells with TLR4 as a therapeutic target. We now changed the sentence accordingly (changes highlighted in red): In summary, we determined that ME-CSCs mediate the inflammatory environment of cholesteatoma via TLR4-mediated NF-kB-signaling, suggesting a distinct role of ME-CSCs as drivers of cholesteatoma progression and TLR4 on ME-CSCs as a therapeutic target. We were not able to include the suggestion made by the referee due to word limitations in the abstract.

Introduction: In summary, we were able to determine ME-CSCs as cellular mediators of the inflammatory environment of cholesteatoma via TLR4-mediated NF-kB-signaling. This is another confusing sentence, because I thought your main idea is ME-CSCs, which were stimulated by inflammation, form and progress cholesteatoma. Can you make it more clear regarding your findings?

We now rephrased the sentence as follows: “In summary, we determined that ME-CSCs regulate the inflammatory environment of cholesteatoma via TLR4-mediated NF-kB-signaling, suggesting a distinct role of ME-CSCs as drivers of cholesteatoma progression in an inflammation-dependent manner.”

Results:

Figure 1: figures were good. However, it will be optional if you can add stem cell fluorescent pictures overlapping with common and different stem cell markers to differentiate ME-CSCs and ACSCs.

Thank you for this remark. In a previous study (Scientific Reports, (2018) 8:6204), we identified and characterized ME-CSCs and ACSCs in great detail, including a range of marker panels as well as  epidermal stem cell characteristics sphere formation, self-renewal and multilineage differentiation. Most theories describing cholesteatoma formation assume a disturbance of the tympanic membrane resulting in deposition of auditory canal tissue into the middle ear, which might also comprise the predecessor cells of ME-CSC. Since these observations are already published an available to the reader, did not include any analysis of stem cell markers in the present manuscript. In addition, ME-CSCs and ACSCs share their distinct marker panel and the epidermal stem cell characteristics described above and are only distinguishable in terms of their pathogenic phenotype. In this regard, we showed that exposure of ME-CSCs to factors mimicking the microenvironment of the cholesteatoma (KGF, EGF, HGF and IGF-II) resulted in differentiation into keratinocyte-like cells, which was not observable in ACSCs (Scientific Reports, (2018) 8:6204). We therefore concluded that ACSCs are suitable control cells for ME-CSC, since they share certain epidermal stem cells characteristics, but lack the pathogenic phenotype of ME-CSCs.

Figure 3. If translocation of NF-KB p65 is induced by LPS, you should show your readers that the use of LPS-RS can reverse the translocation. Personally, I am so curious the TNF-alpha may cause the translocation?  If so, the figure 7 may be more intact and complete.

We thank the reviewer for raising this issue. LPS-RS was already reported to prevent translocation of NF-kB into nucleus of microglia (Gaikwad et al., Int J Inflam. 2015; 2015: 361326.), providing a proof-of-principle that LPS-RS block NF-kB-activity. In this line, we observed strongly reduced expression levels of NF-kB target genes in ME-CSCs after application of LP-RS. We thus suggest LPS-RS to prevent nuclear translocation also in ME-CSCs, although the very short time for addressing the reviewer’s comments (10 days) did not allow us to address this matter on an experimental level.

Have your ever tried TLR4 inhibitor Sulforaphane (SFN) [1-isothiocyanato-4-(methylsulfinyl)butane]? Would effects of SFN are similar or different to LPS-RS?

We thank the reviewer for these interesting questions. We did not apply other inhibitors of TLR4 signaling like Sulforaphane in the present study, since Eritoran was already applied in different clinical trials against sepsis e.g. phase II (J Infect Dis, 180 (5), 1584-9) and even phase III (JAMA 309 (11), 1154-62). We now included this aspect in the discussion section (lines 418-420). We can only speculate regarding the effects of SFN on ME-CSCs, but suggest SFN to similarly inhibit TLR-4 signaling in ME-CSCs. We thank the reviewer for suggesting SFN as a further inhibitor of TLR4 signaling next to Eritoran, which led us to including SFN into the discussion section (lines 413-15).

Discussion:  too many redundant and repeated words, like many “accordingly”….

We now changed redundant and repeated words, like “accordingly” in the revised version of the manuscript.

Some syntax errors in the content need to be revised.

We carefully revised the manuscript and corrected syntax errors. Furthermore, we corrected content towards a more streamlined introduction and discussion.